# Cultivars Resistance Assay for Maize Late Wilt Disease

**DOI:** 10.3390/biology11121854

**Published:** 2022-12-19

**Authors:** Ofir Degani, Ran Yifa, Asaf Gordani, Paz Becher, Assaf Chen

**Affiliations:** 1Migal—Galilee Research Institute, Tarshish 2, Kiryat Shmona 1101600, Israel; 2Faculty of Sciences, Tel-Hai College, Upper Galilee, Tel-Hai 1220800, Israel; 3CTS Group, Seed Department, Kiryat Malakhi 8305769, Israel

**Keywords:** *Cephalosporium maydis*, crop protection, disease control, fungus, *Harpophora maydis*, late wilt, *Magnaporthiopsis maydis*, maize, pathogenicity, real-time PCR

## Abstract

**Simple Summary:**

Maize late wilt disease (LWD) is considered the most severe corn disease in Israel and Egypt and a significant threat in other countries. The utilization of disease-resistant maize cultivars is currently our best LWD control method. Here we evaluated the predictive ability of a rapid test to examine the susceptibility/tolerance of corn varieties to the disease. The fastest assay is based on the sensitivity of corn seeds to metabolites secreted by the pathogen. However, it is only reliable in detecting highly resistant or extremely susceptible maize varieties. To identify LWD immunity of mildly tolerant hybrid plants, a full-season pot assay in the open air, under field conditions, is needed. New detection methods, including aerial visible dehydration symptom monitoring and real-time PCR-based tracking of *M. maydis* DNA inside the plants’ tissues, were evaluated here and proved to provide a great advantage. No complete overlap exists between the fungal DNA amount and the severity of symptoms. Such a correlation exists in high sensitivity or resistance cases but not in intermediate situations. Still, the valuation of the pathogen’s establishment in asymptomatic corn hybrids is a powerful means to predict each maize variety’s LWD immunity.

**Abstract:**

*Magnaporthiopsis maydis* late wilt disease (LWD) in corn is considered to be the most severe in Israel and Egypt and poses a significant threat in other countries. Research efforts extending over a period of five decades led to the development of chemical, biological, agrotechnical, physical (solar disinfection) and other means for controlling late wilt disease. Today, some applications can reduce damage even in severe cases. However, cultivating disease-resistant maize varieties is the primary means for reducing the disease’s impact. The current work uses a rapid (six days) laboratory seedling pathogenicity test and a full-season open encloser semi-field conditioned pots assay (101 days) to classify maize varieties according to their LWD resistance. To better evaluate differences between the cultivars, a real-time based molecular assay was applied to track the pathogen’s presence in the plants’ tissues, and visible light aerial imaging was used in parallel. The findings show that in cases of extreme sensitivity or tolerance (for example, in the highly susceptible Megaton cultivar (cv.) or the resistant Hatai cv.), a similarity in the results exists between the different methods. Thus, a reliable estimate of the varieties’ sensitivity can be obtained in a seed assay without the need for a test carried out throughout an entire growing season. At the same time, in most situations of partial or reduced LWD sensitivity/resistance, there is no match between the various tests, and only the entire growing season can provide the most reliable results. Tracking the amount of *M. maydis* DNA in the plants’ bodies is a precise, sensitive scientific tool of great importance for studying the development of the disease and the factors affecting it. Yet, no complete overlap exists between the fungal DNA amount and symptom severity. Such a correlation exists in high sensitivity or resistance cases but not in intermediate situations. Still, the valuation of the pathogen’s establishment in asymptomatic corn hybrids can indicate the degree of LWD immunity and the chance of susceptibility development.

## 1. Introduction

The pathogen *Magnaporthiopsis maydis* (former names *Harpophora maydis* and *Cephalosporium maydis*) [1,2,3,4,5] causes severe damage to corn fields in the late stages of growth (near the time of harvest) [6,7]. This disease, known as maize late wilt disease (LWD), has spread since its discovery (in the 1980s) and is now common in most corn-growing areas in Israel [8]. A similar scenario is prevalent in other countries, and the areas highly impacted by LWD now include Egypt [9,10,11,12,13], India [14,15,16,17], Portugal and Spain [18,19,20,21,22]. In some fields and in sensitive plant species, the disease can affect 100% of the plants [23]. However, most LWD outbreaks have resulted in 40–50% economic losses [24,25].

The disease is characterized by a relatively rapid wilting of 60–80-day-old corn, from before the flowering stage (tasseling) to physiological ripening [12,26,27,28]. The first signs of dehydration may appear 50 days post sowing, progress from the lower part of the plant upwards, and eventually cause dehydration and damage to the cobs [21]. *M. maydis* is a hemibiotroph, seed-borne [29] and soil-borne [30] pathogen. It can survive as an endophyte (living inside the plants asymptomatically or with minor symptoms) in LWD-resistant corn and secondary host plants [26]. These include green bristle (*Setaria viridis*, green foxtail), watermelon (*Citrullus lanatus*), cotton (*Gossypium hirsutum* L.) and lupine (*Lupinus termis* L.) [31,32]. As such, it may also be an opportunist (“seizing opportunities”) and attack the hosts (corn or cotton) in special cases, when conditions are favorable [26].

Many control methods have been tested against the LWD causal agent over the years (recently [9,33,34,35,36]). A focused research effort in recent years has yielded success, and now there is an effective and economical chemical (Azoxystrobin-based) fungicide method for dealing with the disease [23,37]. At the same time, application of this method requires changes in the growth method (double-row cultivation and dripline irrigation) [38], and there is a constant fear of the development of fungal resistance against the preparations [39]. In addition, such chemical interventions over time are accompanied by environmental impact (such as harming beneficial microorganisms in the soil) and health risks [40]. One research direction designed to address these challenges is the utilization of biological pesticides. The potential for applying natural pesticides against *M. maydis* lines in Israel has been tested with different degrees of success in recent years [37]. Together with experiments carried out abroad [11,35,36,41,42,43,44,45,46,47,48,49], these efforts indicate a clear advantage for developing such an application [50].

Today, the main means of dealing with LWD is the cultivation of disease-resistant corn varieties [10,13,14,15,16,17,51,52,53,54]. This strategy is eco-friendly and cost-effective to implement but requires constant efforts to screen and identify new resistant corn hybrids. The reason is that prolonged growth of a resistant maize cultivar (for several years) may eventually result in selection pressure that will lead to the outbreak of virulent strains of the pathogen capable of causing disease, as previously reported in Egypt and Spain [19,20,55]. This scenario occurred in Israel for the relatively resistant corn variety, Royalty, which became the leading corn variety during the LWD outbreak in the 1990s [26,37]. A program to develop new corn hybrids resistant to LWD has been operating in Egypt since 1980 [56] and in Israel for more than a decade (R&D North, Migal—Galilee Research Institute, Kiryat Shmona, Israel) [8], and has recently also been reported in India [16].

These LWD resistance trials are commonly based on growing corn varieties in cornfields heavily infested with the *M. maydis* pathogen (Israel and Egypt). They also rely on inoculating plants by injecting the pathogen directly into the stem and estimating the amount of browning generated as observed in a longitudinal section of the stalk (Egypt and India) [15]. Newer measures are based on field remote sensing to detect dehydration symptoms in the cultivars tested (Spain and Israel). Remote sensing visible and thermal imaging (using an unmanned aerial vehicle—UAV) of the crop canopy allows a large number of plants to be scanned over an extensive area. This method was previously tested in Spain to optimize irrigation [19] and has been applied in Israel to detect LWD early wilting symptoms and evaluate the effectiveness of preventative treatments [38]. The latest report has proven that thermal sensing is reliable and accurate and is able to detect tiny changes in the landscape temperatures of the plants measured (with an accuracy of less than 1 °C). In 2019, the remote sensing method was applied to study the effect of *M. maydis* in fodder corn plants [8]. The experiments were conducted as part of an annual cultivar test in a commercial field with a long history of LWD infection. Visible RGB (red, green, blue) imaging was used to create the green-red vegetation index (GRVI). These measurements correlated the plant’s health status or thermal imaging during the growth season. They also paralleled the disease progression evaluated by molecular monitoring of the pathogen’s DNA inside the host plants and the plants’ growth parameters and yield at the season’s end. A drawback of these measures to assess maize cultivars’ LWD resistance is that they require growing the plants during a full growing season under field conditions. Hence, they require a significant financial investment, are time-consuming, and depend on the level of infestation of the selected field, the spread of the disease in the area and the unstable environmental conditions.

Despite significant research carried out in recent decades to identify the causes of LWD in resistant corn varieties, there is still more hidden than known knowledge about such genetic causes. The accumulating evidence shows that several genes are involved in determining disease-resistant traits [51]. In addition, resistance to the disease is associated with chemical and histological changes in plant tissues [57]. Among other things, resistant plants had a higher phenolic and lower sugar content, which allowed the plants to cope better with the pathogen. Moreover, changes in the tissues surrounding the xylem that may create a mechanical barrier against *M. maydis* invasion have been documented in resistant plants [57]. More vascular xylem tissues were found in resistant plants, facilitating water flow and overcoming blockage created by the pathogen. Finally, plant hormones such as auxin and cytokinin have been shown to inhibit the pathogen [26]. Differences in the balance of these hormones may play a role in the resistance of corn varieties to the disease.

The current work aims at evaluating a rapid method (six days) for testing the resistance of corn varieties to LWD based on in vitro seed pathogenicity and comparing it to a pot assay under an open-air enclosure throughout a complete growing season (ca. 100 days). To this end, 12 corn hybrids with varying susceptibility to the disease were selected. In the pots assay, apart from growth evaluation and symptom estimation, the disease impact was evaluated using drone (UAV) photography from the air and determining the pathogen’s DNA amount in the plant tissues (based on the real-time PCR method).

## 2. Materials and Methods

### 2.1. The Maize Varieties Selection and Experimental Design

The assays were conducted as blind tests (except for the controls), so each maize cultivar received a number. The cultivar’s name was only revealed after collecting and analyzing the data. Nine selected fodder varieties were transferred by the CTS Group, Tel Aviv, Israel (Table 1). Two sweet corn varieties used as control are sensitive to LWD—Megaton cv. and Prelude cv. [58]. Additionally, two fodder corn varieties were utilized as controls: the Hatai cv. (LWD-resistant) and P32W86 cv. (W86, LWD-susceptible). The Hatai cv. was tested twice as an internal control (once blinded as cultivar number eight and once unblinded).

### 2.2. M. maydis Origin and Growth Conditions

One representative isolate of *M. maydis* called *Hm-2* (CBS 133165) was chosen for this study. This isolate was deposited in the CBS-KNAW Fungal Biodiversity Center, Utrecht, The Netherlands, after being characterized by its pathogenicity, physiology, colony morphology, and microscopic and molecular traits [59]. The growth of *M. maydis* in a solid potato dextrose agar (PDA; Difco Laboratories, Detroit, MI, USA) medium was carried out by transferring 6-mm-diameter disks from the edges of a young culture that had grown for 4–6 days to a new growth plate. The plates were incubated at 28 ± 1 °C in the dark. For the growth of mycelium in a liquid medium, five such discs were inserted into an Erlenmeyer flask with 150 mL of potato dextrose broth (PDB; Difco Laboratories, Detroit, MI, USA). The cultures were incubated for 10 days in a shaking incubator at 150 rpm at 28 ± 1 °C in the dark.

### 2.3. In Vitro Seed Test

The *M. maydis* isolate *Hm-2* was grown in a liquid medium for 10 days, as described above. The liquid growth medium was filtered using a Buchner funnel with double Whatman paper, centrifuged at 6000 rpm for 20 min and further filtered through a 0.4-micron membrane. Seeds of the tested corn and control varieties (Table 1) were washed in 1% sodium hypochlorite (NaOCl) and sterilized DDW under sterile conditions. They were then immersed in the *M. maydis* culture filtrate for six hours. Seed of the control varieties was immersed in PDB medium or sterile tap water. The seeds were then dried and transferred to Petri dishes with wet Whatman paper soaked with sterilized tap water and incubated at 28 ± 1 °C in the dark for a week. Each group of seeds was tested in 10 biological repetitions (repetition is a Petri plate containing 10 seeds). Germination percentages were tested every two days for up to six days. At the experiment’s end, the wet weight of the germinating seeds was checked.

### 2.4. Pots Assay under Semi-Field Conditions throughout an Entire Growing Season

#### 2.4.1. The Experiment Conditions

In this test, carried out at the North R&D plantation farm (Hula Valley, Upper Galilee, northern Israel, 33°09′08.2″ N 35°37′21.6″ E), nine selected fodder corn varieties were tested against three plants of the control varieties (listed above, Table 1). The assay was carried out in five repetitions per treatment (each repetition was a 10 L pot containing five sprouts). Pots were filled with a heavy local soil from the farm with no history of LWD. If such an infestation occurred, it was expected to be minor. The soil was mixed with 30% coarse perlite for aeration. Inoculation was performed by mixing the soil’s upper part with sterilized infected wheat grains (150 g incubated with *M. maydis* for about 10 days at 28 ± 1 °C in the dark). This procedure took place one week before sowing. With sowing, a complementary soil infection was conducted by adding 3 *M. maydis* colony mycelia discs (see Section 2.2) to each maize seed. The control pots underwent similar treatment but without the pathogen.

Computerized irrigation was carried out with driplines once every two days, with an amount adjusted to maintain moderate humidity conditions (usually 2 L per pot/day). Throughout the season, fertilizers and treatments against various pests were applied to keep the disease factor only from the *M. maydis* infection.

#### 2.4.2. Growth and Disease Estimation

After one week, the soil-surface peek percentages were checked in each treatment. On day 50 (the V5-V6 plants’ growth stage, about 10 days from the fertilization stage—R1) and day 101 (in the milk ripening stage—R5), the appearance of symptoms, the phenological phase, and the plants’ wet weight and height were measured. The cobs yield (fresh weight) was determined at the harvest. Additionally, the presence of the fungus DNA in the root (day 50) or in the lower stem (day 101) of each plant was quantified by qPCR. At the season’s end, the total health status of the plants was evaluated based on four categories: 1—healthy, 2—symptoms, 3—diseased, and 4—dead. Visible symptom evaluation was performed in parallel from the air and from the ground.

While aerial imagery allows easy scanning of a large group of plants and identifies wilted plants and “hot spots of disease” in the field, this method is less accurate than evaluating the symptoms from the ground. On the other hand, symptom evaluation from the ground requires investment in time and workforce and, if conducted by several persons, can be subjected to differences in individual judgment. We used both ways to demonstrate their advantages and weak points so that this information can be considered and assist in decision-making regarding the preferred evaluation method of choice in future works.

### 2.5. Molecular Real-Time PCR Diagnostic

#### 2.5.1. DNA Extraction

The plants’ parts were washed thoroughly with running tap water, then twice with sterile double-distilled water (DDW) and sliced into ca. 2 cm sections. The total weight of each repeat was adjusted to 0.7 g. The pathogen’s DNA was isolated and extracted according to a previously published protocol [60] with slight modifications [59].

#### 2.5.2. qPCR Technique

The quantitative real-time PCR method was performed on an Applied Biosystems (Foster City, CA, USA) ABI-7900HT device (384 well plates). The technique relies on a standard qPCR protocol, which detects mRNA (cDNA), but is optimized to detect *M. maydis* DNA [23]. The qPCR reaction is performed in a total volume of 5 μL per reaction (with 4 repetitions): 0.25 μL of each primer (Forward/Backward at a concentration of 10 μM), 2.5 μL of a ready reaction mixture—iTaq™ Universal SYBR Green Supermix solution (Bio-Rad Laboratories Ltd., Hercules, CA, USA) mix and 2 μL DNA template. Reaction conditions: 95 °C for 60 s, 40 cycles of 95 °C for 15 s, 60 °C for 30 s, and finally creating a melting curve.

The A200 primers amplify the segment-specific to *M. maydis*:

Forward—5′-CCGACGCCTAAAATACAGGA-3′

Backward—5′-GGGCTTTTTAGGGCCTTTTT-3′

The primers for the COX gene (codes for the enzyme cytochrome oxidase, the last enzyme in the cellular respiratory chain in the mitochondria) that is used as a housekeeping gene to normalize *M. maydis* DNA amount according to the ΔCt model:

Forward—5′-GTATGCCACGTCGCATTCCAGA-3′

Backward—5′-CAACTACGGATATATAAGRRRCCRRAACTG-3′

### 2.6. Statistical Analysis

The in vitro seed assay and the semi-field pots evaluation were analyzed using the same statistical method with JMP software, version 15 (SAS Institute Inc., Cary, NC, USA). The differences in results were calculated using one-way analysis of variance (ANOVA) and Student’s t-test post hoc (without multiple tests correction, *p* < 0.05).

## 3. Results

### 3.1. In Vitro Seed Test

The in vitro seed assay is aimed at rapidly screening resistance of maize cultivars to late wilt disease (LWD). The sensitivity of seeds to the *M. maydis’* metabolites (culture filtrate) was previously demonstrated to serve as a means to study the pathogen’s aggressiveness toward maize [59,61]. However, until now, as far as we know, it was never used to test the corn hybrids’ LWD tolerance. Using pure PDB substrate (without *M. maydis* growth) causes an osmotic effect that inhibits corn seed germination. Yet, after four days, most corn hybrids reached about 100% germination in this substrate (Figure 1, graph insert).

Two corn varieties, Laurca (variety 6) and Megaton, showed limited germination in pure PDB medium (26%) even after six days. Even so, it should be noted that the growth medium content changed during the fungal growth, so the new PDB medium differs in its ingredients from the post-growth (used) medium, i.e., the PDB medium is an imperfect control.

Soaking maize grains of all cultivars in the fungal filtrate for 6 h before seeding in Petri dishes inhibited their germination. In the pathogen’s growth medium, the different cultivars’ seeds showed variable sensitivity (Figure 1). Variety 1 (KXB9571), Megaton and W86 (P32W86) did not germinate in the six days of the test. Varieties 6 (Laurca), 8 (Hatai) and Prelude germinated at the rates of 3%, 17% and 33%, respectively. Corn varieties 4 (Leonidos), 7 (Colossus) and 9 (32D99) showed a higher germination capacity (62%, 63% and 72%, respectively). Finally, varieties 2 (KAC0572), 3 (KES Elektro) and 5 (Calumet) were the most resistant to the influence of the fungal growth culture filtrate.

Monitoring the wet weight of the germinating seeds after six days (Figure 2) in *M. maydis*’ culture filtrate shows that varieties 4 (Leonidos), 7 (Colossus), 8 (Hatai), 9 (32D99) and W86 had low values. Especially the disease-sensitive variety, Megaton, was impacted. Its seed biomass was about 50% lower than the other seeds (probably due to its germination inhibition, Figure 1). On the other hand, varieties 2 (KAC0572), 3 (KES Elektro) and Prelude achieved the highest values.

### 3.2. Pot Assay under Semi-Field Conditions throughout an Entire Growing Season

#### 3.2.1. Aboveground Appearance Rate Estimate

Resistance of the fodder corn varieties to LWD was evaluated throughout a full growing season in pots under semi-field conditions against control varieties. The structure of the experiment and percentages of emergence after one week are described in Figure 3. Emergence rates in the soil of hybrids 4 (Leonidos) and W86 were significantly lower compared to the other cultivars. This was also true for variety 6 (Laurca), but this could be related to its overall low germination capacity (Figure 1, insert) and therefore should not be attributed to the disease. A significant difference compared to the control (natural tilled soil without complementary inoculation) was found in varieties 3 (KES Elektro), 4 (Leonidos), 7 (Colossus) and Megaton (*p* < 0.05, *t*-test). In these varieties, the above-soil sprout rate in the infected soil decreased by about 40–50%. The hybrids Hatai, 5 (Calumet) and 9 (32D99) stood out favorably in this index, without significant damage to the emergence in the pathogen-enriched soil.

#### 3.2.2. Growth Indices on Day 50 of Growth

Growth indices of the corn varieties were evaluated on day 50 from sowing (Figure 4). Trends obtained in the wet weight and height indices are similar. Variety 6 (Laurca) stood out as having a high value in these indices, with hybrids 5 (Calumet), 7 (Colossus), 9 (32D99) and Megaton below it (without a significant difference). Relatively low values were measured for varieties 3 (KES Elektro), 4 (Leonidos), and, to a less extent, cultivar 8 (Hatai). The phenological maturity of the corn varieties evaluated at this age according to the number of leaves shows that variety 2 (KAC0572) was the most developed. In contrast, cultivars 1 (KXB9571), 4 (Leonidos), and 9 (32D99) were the least developed (*p* < 0.05).

#### 3.2.3. Infection Level throughout the Growing Season

The amount of the pathogen’s DNA in the roots of the plants on 50 days after sowing (Figure 5) does not correspond to the changes in growth indices. Still, it is consistent with what is known about some of the previously tested varieties. For example, the susceptible W86 and hybrid 7 (Colossus) [8,23] showed drastically higher values (ca. 500% and 450%) than the control. High DNA levels of the pathogen were also found in variety 6 (Laurca), which was considered resistant until now [8]. DNA values were lower but still higher than the uninfected control in varieties with prominent growth retardation symptoms, 3 (KES Elektro) and 4 (Leonidos). In the wilted Megaton cv., minor DNA values were measured, apparently as a result of the disintegration of the fungal hyphae in those severely affected plants. At the end of the growing season (day 101), the DNA indices increased in general, with a sharp jump in the amount of DNA of the pathogen in varieties 4 (Leonidos) and 5 (Calumet) that were infected (to 120.5 and 111.0-fold higher compared to the control). At this sampling day (101), high DNA values were measured in decreasing order in cultivars W86, 9 (32D99), 7 (Colossus), 6 (Laurca), and 3 (KES Elektro). In varieties 1 (KXB9571) and 2 (KAC0572), zero values were measured on both sampling days.

In highly susceptible varieties such as W86 cv. and Megaton cv. plants, the high infection levels (*M. maydis* DNA in the stem) at the harvest (day 101 from sowing, 41 days from fertilization) were accompanied by an increase in drying out symptoms. Those included the color alteration of the exterior surface of the lower stalks. A cross-section near the first aboveground internode of the diseased maize plant (Figure 6A,B) reveals the tissue’s yellow-brown color. This contrasts with the tissue’s green color in a healthy plant cross-section (Figure 6C). Also, the pathogen presence and spread led to vascular bundle occlusion, which appears as brown spots.

The experimental images from the ground and from the air are shown in Figure 7. Evaluation of the severity of the symptoms using visible aerial imaging near the season’s end (Figure 8) proved effective and supported previously demonstrated results [8].

#### 3.2.4. Growth Indices and Dehydration Level at the Harvest

At the time of harvest, 101 days after sowing, various measurements were made to evaluate differences between the cultivars. The experiment was imaged from the air, and the dehydration estimation of the plants’ landscape was assessed from the images’ analysis (Figure 8). The number of diseased plants was high in corn varieties 5 (Calumet), 9 (32D99), and Megaton, and low in varieties 7 (Colossus) and Hatai (both repeats). A statistically significant difference (*p* < 0.05) was measured only for the Megaton cv. The percentage of diseased plants compared to the control shows a certain overlap with the estimate above, with high values in the Megaton, 3 (KES Elektro) and 9 (32D99) varieties, and low values in the 7 (Colossus), 1 (KXB 9571) and Hatai. Hybrids 7 and Hatai were also evaluated as being resistant in the symptoms’ estimation from the ground (Table 2). Still, unlike what was expected, variety 9 (32D99), classified as being susceptible in the aerial imaging, was evaluated as being resistant in the ground inspection. In fact, the ground estimation supported the previous reports that this variety is LWD immune [8,23]. In assessing dehydration from the ground, variety 4 (Leonidos) was classified as being LWD-resistant with only 32% wilting. On the other hand, varieties 5 (Calumet) and Megaton showed 60–68% dehydration.

A growth estimation conducted at harvest (Table 2) showed some parallels between the development of the plants and the disease symptoms. Such a correlation was found in variety 4 (Leonidos), which exhibited low wilt signs. In this variety, the plant and cobs wet weight values were evidently higher than the control (soils without the complementary inoculation). On the other hand, the growth indices were significantly impaired in the 5 (Calumet), W86 and Megaton corn varieties that showed severe symptoms.

Overall results of the tests and the corn varieties’ approximate resistance order according to integration between them are shown in Table 3. The table summarizes the results of the seed and pot assays and presents the LWD resistance rank that each maize variety achieved in those tests. This comparative analysis illustrates that in severe cases of disease (for example, in the sweet corn Megaton cv.), there is a parallel in the results between the different methods, and a reliable estimate of the varieties’ sensitivity can be obtained without the need for an entire growing season test. Calculating the sum score without the seedling phase (in vitro sprouts germination and pots aboveground emergence) resulted in the same cultivar LWD resistance order, except for variety 4 (Leonidos), which was re-ordered as less tolerant after varieties 2 (KAC0572) and 1 (KXB 9571).

At the same time, in most cases of partial or reduced sensitivity/resistance, no agreement exists between the in vitro seed tests and pot assay. Thus, there is no escaping conducting a full growing season that provides the most reliable data. Evaluating the *M. maydis* DNA amount in the plants’ body is a highly sensitive scientific tool of great importance for studying the development of the disease and the factors affecting it. In cases of high sensitivity or resistance to LWD, a strong correlation exists between pathogen-host colonization and disease severity. At the same time, a lack of complete correspondence between fungal DNA and symptom severity in intermediate situations indicates that host immunity may be reflected not only in eliminating symptoms.

## 4. Discussion

Israel and Egypt are considered among the world’s most affected areas by *Magnaporthiopsis maydis* maize late wilt disease (LWD) [26]. Research efforts extending over a period of six decades and the development of chemical, biological, agrotechnical, physical (solar disinfection), and other means have produced results. Today, some applications have the potential of dealing with LWD damage [37]. At the same time, the primary means of reducing deterioration of the disease remains the breeding and cultivation of tolerant corn varieties [10]. The causes of resistance are unknown, but evidence points to the involvement of genetic factors, changes in the structure of the tissues, physiological differences, and differences in the dates of ripening and flowering as being related to the resistance/tolerance feature of the corn variety to LWD in corn [13,14,17,51,53,62,63].

Tests for the resistance of corn plants to the disease have been carried out in the past two decades in Egypt [56] and Israel [8], and in recent years also in India [16]. The study presented here suggests using a rapid seeds assay to rule out corn cultivars with strong resistance or susceptibility to the disease. Since sensitivity during germination does not coincide with the symptoms that will appear later in the growth, it must be taken into account that germination is affected by other aspects besides harmful metabolites secreted by the fungus [61]. These can be the components that cause osmotic or ionic stress. Hence, this subject clearly requires further investigation in future research.

Germination inhibition in the culture filtrate of *M. maydis* (Figure 1) was not always linked to the seedlings’ initial growth fresh weight values (Figure 2). The pathogen is not necessarily the sole reason for the sprouts’ low growth indexes. For instance, Prelude cv., which hardly germinated in the growth liquid of the pathogen, had significantly higher biomass than most cultivars tested. Still, there is a parallel in Megaton cv. (and, to a lesser extent, Colossus cv.) between the rate of germination and the sprouts’ biomass. Hence, it seems that the sprouts’ biomass is affected by factors that are not only related to the disease (such as the vitality of the seeds, their carbohydrate content and the genetic differences between them).

An entire season trial (about 80–100 days) must be executed to identify the resistance level of other less prominent situations. Yet, field environments are unstable, and their variables (soil composition and microflora, watering level, *M. maydis* infection level and dispersal, etc.) cannot be monitored or determined. Therefore, other approaches are preferred, such as the semi-field pots assay presented here.

It arises from the current study and others [8] that LWD tolerance may be unstable and could be altered in different whole-season circumstances. For example, Laurca cv. (variety 6), which up until now was thought to be LWD-resistant, was classified in the current study as being moderately susceptible. This change can be due to immunity compromising. A previous evaluation of maize cultivars’ susceptibility to LWD [8] found that even the highly resistant 32D99 cv. (variety 9) had suffered from severe LWD symptoms in some plants. In contrast, no apparent disease symptoms could be identified in some plants of the highly susceptible L.G.30.669 genotype.

So far, there is no clear explanation for the disease appearance in resistant cultivars or the unexpected healthy plants of a susceptible cultivar. Are these differences the consequences of variations in the host, the causal disease agent, or the surroundings? Some lines of evidence state that such host immunity instability may result from pathogenic variations in *M. maydis* itself since highly violent lines of the pathogen were discovered in Egypt [55] and Spain [20]. New information from a net-house full-season study proved that highly pathogenic lines of the pathogen also exist in Israel [64].

One previous study [65] provides a vital clue to the question: what are the differences between these fungal lines? The researchers reported that aggressive *M. maydis* isolates showed lower β-glucosidase, cytochrome oxidase, polyphenol oxidase activity and peroxidase enzymes, and higher catalase dehydrogenase activity than less pathogenic isolates [65]. Further work is essential for expanding this information to provide us with a better understanding of the LWD pathogen.

Regarding the growth indexes, natural differences undoubtedly exist between corn cultivars. This being said, it is most certain that the late wilt pathogen has an effect in some cases. For example, the fodder cultivar 4 (Leonidos), which sprouted well in plates (Figure 1, insert), was severely affected by the *M. maydis* hostile presence. It had reduced seed germination in the pathogen medium filtrate, slow germination and initial development in pots, and significantly lower growth indices 50 days after sowing. Yet this cultivar recovered toward the season-ending, and its high growth indexes and health state positioned it as one of the most resistant corn varieties to the disease. The high pathogen levels within cultivar 4 (Leonidos) tissues may explain some of the above changes and reveal how fragile the immunity of this species is.

State-of-the-art measures based on field remote sensing and molecular monitoring of the pathogen’s DNA inside the host tissues are becoming increasingly significant. High levels of the pathogen (DNA qPCR detecting) within asymptomatic corn varieties may suggest a greater future risk of LWD immunity compromising in those varieties in the future. It may also point to the possibility that some specific hybrids’ colonization affects *M. maydis*’ pathogenic behavior, which tends to be more biotrophic than necrotrophic, i.e., the pathogen preserves itself as an endophyte in those corn hybrids until favorable circumstances allow a more violent outburst. The molecular detection of the infection, regardless of the symptom development, might lead to a valuable application of pre-symptomatic disease controlling methods (such as optimal watering) and decision-making (for example, to advance the harvest date) by maize growers.

Integrating remote sensing means diagnosing the sensitivity/tolerance of the corn varieties to the disease is recommended. A study recently published by us [8] proposes for the first time the use of remote sensing to assess the susceptibility of corn varieties to LWD based on RGB imagery, high-resolution green-red vegetation index (GRVI) and thermal imaging. In a commercial field in Israel’s coastal plain, which has a history of being infected with *M. maydis*, a late wilt sensitivity/tolerance test was conducted for 12 fodder corn varieties. The evaluation of the progress of the disease by sensing from the air, in visible and thermal infrared light, throughout the growing season yielded results that indicate stress in the plants, which are consistent with the molecular detection of the pathogen’s DNA in the plant tissues and the growth and yield evaluation. At the season’s end, this aerial assessment of cultivar susceptibility/tolerance to disease proved important for real-time scanning and evaluation of large plant groups.

In follow-up work, examining acute sprouts’ infection for the cultivars’ LWD resistant assay is recommended. As previously reported [66], such a procedure could be done by stabbing the plants’ lower stem with *M. maydis*-infected toothpicks about three weeks after germination. It would also be interesting to examine the impact of *M. maydis* isolates’ mixture over three periods (before, during and a week after sowing) to create a severe disease that will increase the ability to identify differences between the varieties. The standard inoculation methods used in Israel should be examined against the common protocols in Egypt and India, especially the injection of pathogen spores directly into the stems of the plants and checking the spread of a necrotic spot in the stalk [15]. Also, it would be very interesting to examine the effect of root exudates from resistant and sensitive plants on the development of pathogen colonies, as was done previously [61]. Finally, monitoring additional growth indicators, such as the plant’s water conductivity, the leaves’ chlorophyll contents and the plant’s stress hormones, may improve our understanding of how the plants deal with the disease and identify variations in LWD tolerance between cultivars. See such research directions in [63,67].

## 5. Conclusions

Late wilt disease (LWD) in corn is considered the most serious corn disease in Israel and Egypt and a major threat in other countries. Our primary means to reduce the disease’s impact is the use of disease-resistant maize varieties. The current study aimed at evaluating the predictive ability of a rapid test to examine the susceptibility/tolerance of corn varieties to the disease. This assay is based on the sensitivity of corn seeds to metabolites secreted by the pathogen. While such a test proved reliable in detecting highly resistant or extremely susceptible maize varieties, it failed to predict LWD immunity of other moderate hybrids. The advantages of testing potted plants in the open air under field conditions are the management and timing of irrigation, the integration of control treatments without contamination, and achieving greater uniformity in the soil conditions. Such full-season trials are today the only consistent and reliable way to determine the cultivars’ LWD resistance. New detection methods such as aerial visible and thermal dehydration symptoms monitoring and real-time PCR-based tracking of *M. maydis* DNA in the plants can provide a great advantage. They can allow scanning large portions of the field by sensitively monitoring tiny differences in the plants’ canopy temperature. The molecular method, on the other hand, enables us to closely monitor the pathogen’s establishment and spread within the plants’ tissues. Together, these techniques could assist in clearly classifying each maize variety’s LWD immunity.

## Figures and Tables

**Figure 1 biology-11-01854-f001:**
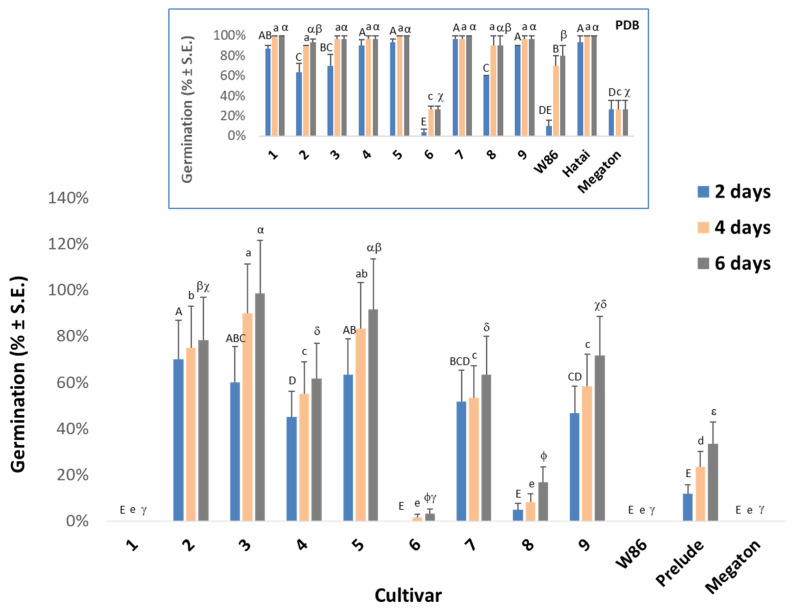
Seed pathogenicity assay to evaluate the maize cultivars’ resistance to late wilt disease. The seeds of the maize cultivars tested (Table 1) were kept for 6 h in an *M. maydis* growth medium filtrate (obtained from 7-day-old fungus colonies grown in a potato dextrose broth (PDB) rich medium). The treated seeds were incubated in Petri dishes in humid conditions, at 28 ± 1 °C in the dark. Germination percentages were tested every two days for up to six days. Insert—the cultivars’ seed germination in PDB without the fungus. Vertical upper bars represent the standard error of the mean of 10 repeats (Petri dishes containing 10 seeds). Different letters above the bars represent a significant difference (*p* < 0.05) in an analysis of variance (ANOVA) assay.

**Figure 2 biology-11-01854-f002:**
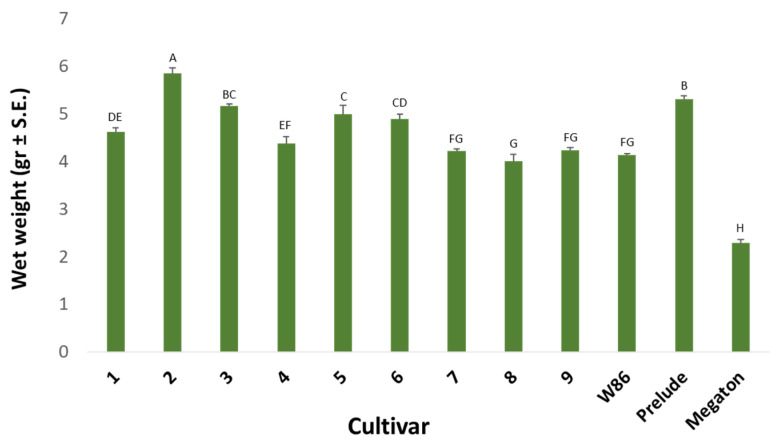
Seedlings’ fresh biomass at day six in the in vitro seed test. The experiment is described in Figure 1. Vertical upper bars represent the standard error of the mean of 10 repeats (Petri dishes containing 10 seeds). Different letters above the bars represent a significant difference (*p* < 0.05) in an analysis of variance (ANOVA) assay.

**Figure 3 biology-11-01854-f003:**
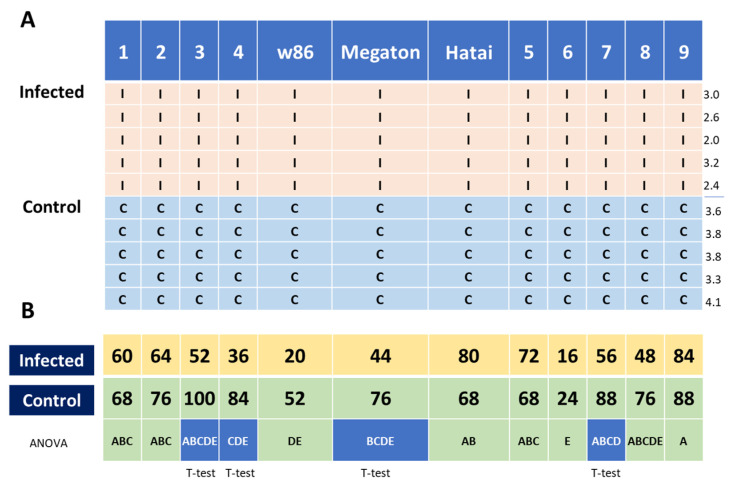
Experiment map (**A**) and above-soil surface appearance evaluation (**B**) in the pots assay under semi-field conditions. The maize cultivars tested are detailed in Table 1. In the experiment map (**A**), the bottom of the scheme faces north. Pink rows are local farm soil inoculated with *M. maydis* (I). Blue rows are control—the same soil without additional infection (C). The numbers on the right are the average sprouts’ emergence in the row, from east to south (per pot out of five seeds). They show that if it exists, a random effect of the experiment design has a minor impact on the results. Sprout aboveground emergence results ((**B**), average percentage per treatment) evaluated after 7 days after sowing. Different letters represent a significant difference (*p* < 0.05) in an analysis of variance (ANOVA) assay. Blue cells differ significantly (*p* < 0.05, *t*-test) from the control treatment in the same cultivar.

**Figure 4 biology-11-01854-f004:**
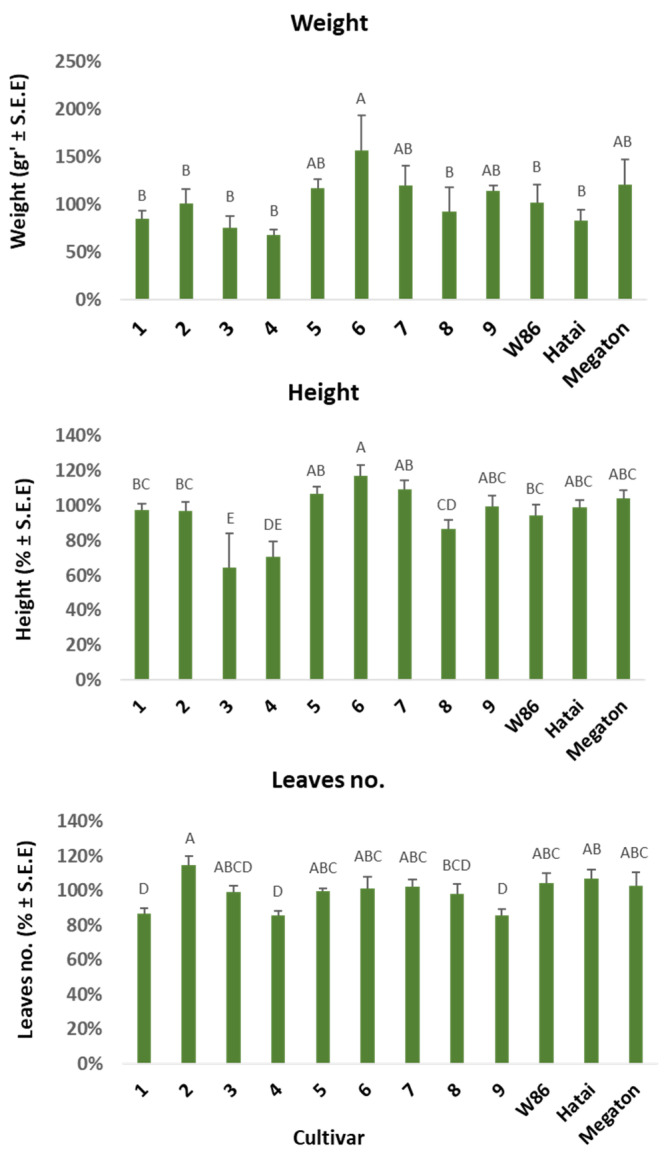
Plants’ growth parameters evaluation 50 days after sowing. The experiment is described in Figure 3. Values represent the average difference (%) of the plants that grew on naturally infected commercial field soil treated with complementary inoculation compared to the control (the same soil without additional inoculation). **Weight**—wet biomass. **Height**—length of aboveground plant parts. **Leaves no**.—the plants’ phenological stage. Vertical upper bars represent the standard error of the mean of five repeats (pots). Values represent percentages different from the control treatment in the same cultivar. Different letters above the bars represent a significant difference (*p* < 0.05) in an analysis of variance (ANOVA) assay.

**Figure 5 biology-11-01854-f005:**
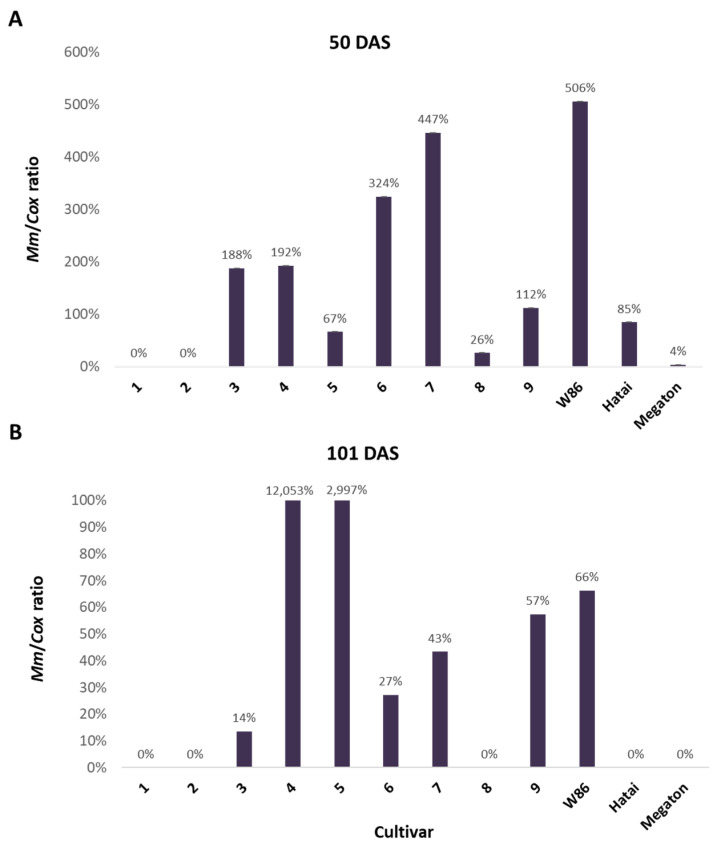
qPCR analysis results on 50 and 101 days after sowing (DAS). The plants in the experiment described in Figure 3 were tested for the presence of *M. maydis* DNA inside their tissues. The *Y*-axis parameters are *M. maydis* relative DNA (*Mm*) levels normalized to the cytochrome c oxidase (*Cox*) DNA. Values indicate the mean of five repeats (pots, each having one plant). Data are the percentages of diseased plants compared to the control treatments (soils without the complementary inoculation). (**A**) *M. maydis* DNA in the plants’ roots at day 50 from sowing. (**B**) DNA in the plants’ lower stem at day 101 from sowing.

**Figure 6 biology-11-01854-f006:**
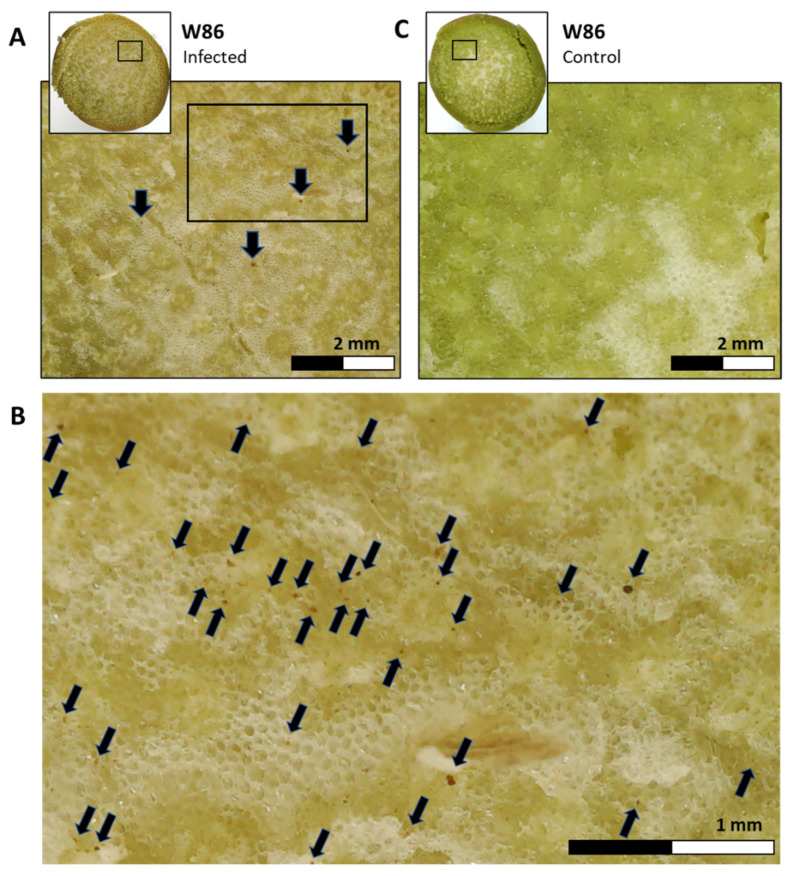
Stem cross-section symptoms in the P32W86 cv. (W86, LWD-susceptible) at the end of the open-air pot experiments (day 101 from sowing, 41 days from fertilization). Cross-section magnification was made of the lower stems (below the first internode) of representative plants from the infection treatment (**A**,**B**) and the non-inoculated control (**C**). The stalk width was ca. 35 mm in diameter. The magnitude area (black square in the whole stalk cross-section, (**A**,**C**)) wide is 7 mm. The width of the magnitude area in (**B**) is 3.5 mm. Late wilt symptomatic plants revealed a color alteration in the lower stems to a yellow-brown hue and vascular bundle occlusions (marked by arrows).

**Figure 7 biology-11-01854-f007:**
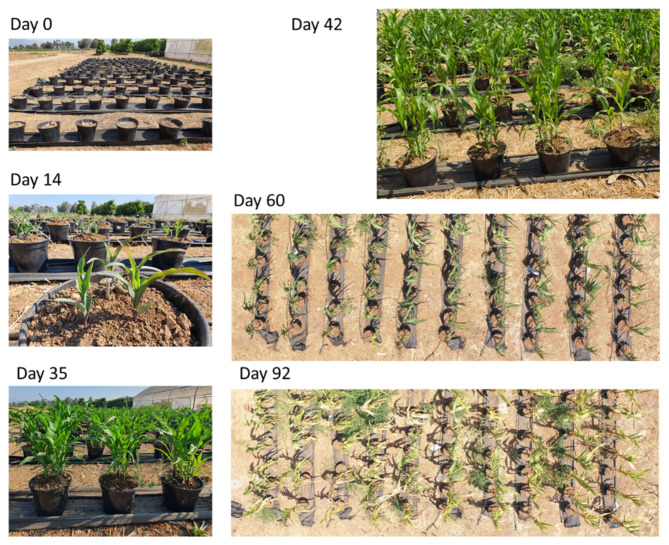
The open-air pots experiment’s pictures. Photos were taken from the ground and air throughout the growing season.

**Figure 8 biology-11-01854-f008:**
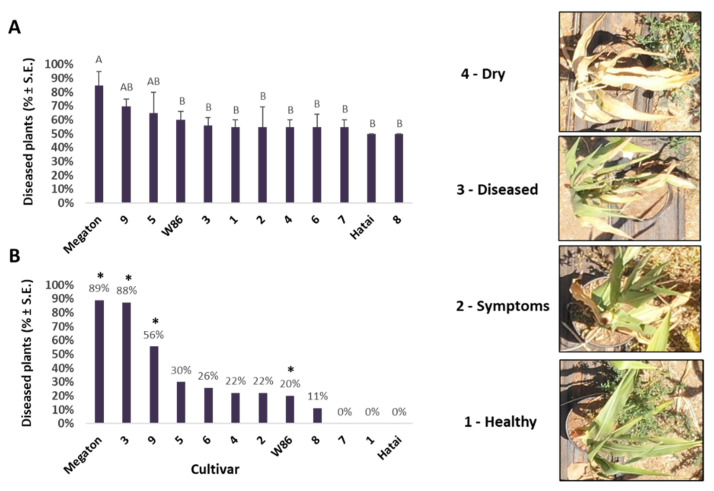
Evaluation of symptom severity from aerial imaging near the season’s end (92 days after sowing). Disease severity was evaluated according to four categories (right panel). The experiment is described in Figure 3. Values indicate the mean of five repeats (pots, each having one plant). (**A**) The percentage of diseased plants in the soil inoculation treatments. Vertical upper bars represent the standard error. Different letters above the bars represent a significant difference (*p* < 0.05) in an analysis of variance (ANOVA) assay. (**B**) The percentages of diseased plants compared to the control treatments (soils without the complementary inoculation). Asterisks represent significant differences (*p* < 0.05, *t*-test) from the control treatment in the same cultivar.

**Table 1 biology-11-01854-t001:** Maize cultivars tested for late wilt disease sensitivity/resistance.

No.	Cultivar	Type	Seed Company	Supply Company	Degree of LWD Sensitivity ^a^
1	KXB9571	Fodder	KWS, Einbeck, Lower Saxony, Germany	CTS Group, Tel Aviv, Israel	N/A
2	KAC0572	Fodder	KWS, Einbeck, Lower Saxony, Germany	CTS Group, Tel Aviv, Israel	N/A
3	KES Elektro	Fodder	KWS, Einbeck, Lower Saxony, Germany	CTS Group, Tel Aviv, Israel	N/A
4	Leonidos	Fodder	KWS, Einbeck, Lower Saxony, Germany	CTS Group, Tel Aviv, Israel	N/A
5	Calumet	Fodder	KWS, Einbeck, Lower Saxony, Germany	CTS Group, Tel Aviv, Israel	N/A
6	Laurca	Fodder	KWS, Einbeck, Lower Saxony, Germany	CTS Group, Tel Aviv, Israel	Resistant [8]
7	Colossus	Fodder	Semillas Fitó, Barcelona, Spain	Tarsis Inc., Petach Tikva, Israel	Sensitive [8,26,59]
8	Hatai	Fodder	Semillas Fitó, Barcelona, Spain	Tarsis Inc., Petach Tikva, Israel	Highly resistant [8,58]
9	32D99	Fodder	Pioneer Hi-Bred International, Inc. Johnston, Iowa, USA	Gadot Agro, Kidron, Israel	Highly resistant [8,23]
10	P32W86 (W86)	Fodder	Pioneer Hi-Bred International, Inc. Johnston, Iowa, USA	Gadot Agro, Kidron, Israel	Sensitive [23]
11	Hatai	Fodder	Semillas Fitó, Barcelona, Spain	Tarsis Inc., Petach Tikva, Israel	Highly resistant [8,58]
12	Megaton	Sweet	Limagrain, Saint-Beauzire, Puy-de-Dôme, France	Hazera Seeds Ltd., Berurim MP Shikmim, Israel	Hypersensitive [58]
13	Prelude	Sweet	SRS—snowy river seeds	Green 2000, Bitan Aharon, Israel	Hypersensitive [23,58]

^a^ The degree of late wilt disease (LWD) sensitivity, as documented in previous works.

**Table 2 biology-11-01854-t002:** Evaluation of the plants’ growth parameters and late wilt symptom severity from the ground at the season’s end (101 days after sowing) ^a^.

Cultivar	DiseasedPlants	Height	Plants’ Wet Weight	Leaves No.	Cobs Fresh Weight
1	48%	99%	114%	90%	76%
2	52%	95%	117%	94%	91%
3	50%	96%	89%	102%	97%
4	32%	93%	125%	95%	159%
5	60%	99%	86%	87%	88%
6	48%	91%	93%	82%	115%
7	44%	98%	135%	100%	102%
8	48%	98%	106%	93%	138%
9	44%	99%	83%	110%	94%
w86	48%	90%	87%	93%	83%
Hatai	44%	94%	124%	108%	128%
Megaton	68%	91%	89%	80%	68%

^a^ Disease severity was evaluated according to four categories. The experiment is described in Figure 3. Values indicate the mean of five repeats (pots, each having one plant). Data are the percentages of diseased plants compared to the control treatments (soils without complementary inoculation). Numbers highlighted in yellow are significantly different (*p* < 0.05, *t*-test) from the control. Numbers highlighted in brown are growth values of 90% and below.

**Table 3 biology-11-01854-t003:** Evaluation of the maize cultivars’ resistance order based on the various tests conducted in this work ^a^.

Seed Assay	Pot Assay						Total Rank
Seed germination 6 days	Sprouts emergence 6 DAS	qPCR 50 DAS	qPCR 101 DAS	Air symptoms 92 DAS	Ground symptoms 101 DAS	Growth indexes 101 DAS	Estimated resistance order
Hatai	Hatai	1	1	Hatai	4	7	7
1	5	2	2	1	Hatai	Hatai	Hatai
5	9	8	8	7	7	2	4
7	1	5	Hatai	8	9	4	2
3	2	Hatai	3	w86	1	8	1
4	w86	9	6	2	6	9	8
9	8	3	7	4	8	1	9
2	6	4	9	6	w86	5	3
8	Meg	6	w86	5	3	3	5
w86	7	7	5	9	2	6	6
6	3	w86	4	3	5	w86	w86
Meg	4	Meg	Meg	Meg	Meg	Meg	Meg

^a^ Values are the cultivars’ number or commercial name (see Table 1) and are ordered top-down according to their LWD resistance. On the top are the highest-resistance cultivars (green cells), the moderate-resistance cultivars are in the middle (blue cells) and the susceptible cultivars are at the bottom (red cells). DAS—days after sowing. The pathogen DNA values are ordered from the smallest value at the top to the highest value at the bottom. The air and ground symptom measures represent the healthy plant percentages. The corn varieties are ordered in the total rank column according to their sum of all scores.

## Data Availability

The datasets generated and/or analyzed during the current study are available from the corresponding author on reasonable request.

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
