# Peer review of "Cultivars Resistance Assay for Maize Late Wilt Disease"

_biology, 2022, doi:10.3390/biology11121854_

Round 1

Reviewer 1 Report

generally in excellent shape and an important contribution.  I found some of the figures difficult to understand.  The authors should work toward greater clarity in the interpretation of the figures.

The photographs could be better explained.

Author Response

Responses to Reviewer 1’s comments

We thank the reviewer for investing substantial efforts, which undoubtedly contribute to this manuscript. The remarks and suggestions improved this paper’s scientific soundness and accuracy. Your contribution is greatly appreciated.

Generally, an excellent shape and an important contribution.

Reply: Thank you for the positive evaluation of our manuscript. All your remarks and suggestions were addressed carefully and thoroughly, as detailed below.

I found some of the figures difficult to understand. The authors should work toward greater clarity in the interpretation of the figures. The photographs could be better explained.

Reply: We did our best to clarify the interpretation of the figures:

  • Figure 1 – we edit the text and the figure legend and replace the figure Y axis title from “emergence” with “germination.”
  • Figure 3 – The figure was edited (A and B were added), and the figure legend was updated accordingly.
  • Figure 5 – we edit the text according to your remark (see details below).
  • Figure 6 – newly added to demonstrate the disease symptoms in the P32W86 cv. (W86, LWD-susceptible) at the end of the open-air pot experiments.
  • Figure 7 – we edit the text.

Replay for reviewer 1’s comments in the PDF file

Line 171

Reply: The sentence was corrected as advised by the reviewer.

Line 180

Reply: The sentence was corrected as advised by the reviewer.

Line 194

Reply: The word “peek“ is correct: “above the soil-surface peek percentages,” i.e., above the soil-surface emergence percentages.

Line 233

Reply: The word “the” was deleted as suggested.

Line 270

Reply: The word “pots” was corrected to “pot,” as advised.

Line 272

Reply: The word “the” was deleted as suggested.

Line 275

Reply: The word “the” was deleted as suggested.

Line 291

Reply: The word “peek” is correct. It means sprout aboveground emergence results. We replace the word “peeking” with the phrase “emergence” for better clarity.

Line 296

Reply: The word “the” was deleted as suggested.

Line 297

Reply: The word “the” was deleted as suggested.

Line 306

Reply: The word “the” was deleted as suggested.

Figure 5B, Are those two high numbers correct? It doesn’t look right.

Reply: Yes, tracking M. maydis DNA variations using the qPCR method can detect variation in proportionate M. maydis specific DNA abundance normalized to the housekeeping cytochrome c oxidase (COX) gene DNA between 10 to 10−6 (more than a million times difference). The actual values here were: 2.4 x 10-6 for the control and 2.9 x 10-4 for variety 4 (Leonidos). So the DNA values in variety 4 were 120.5 fold higher than the control.

This information was added to the text (lines 341-344): “At the end of the growing season (day 101), the DNA indices increased in general, with a sharp jump in the amount of DNA of the pathogen in varieties 4 (Leonidos) and 5 (Calumet) that were infected (to 120.5 and 111.0-fold higher compared to the control).”

Line 393

Reply: The phrase “the pots assay” was corrected to “pot assay,” as suggested.

Table 3, This table is unclear and hard to read.

Reply: You are correct. The following explanation was added to the text to explain this better (lines 428-430): “The table summarizes the results of the seed and pot assays and presents the LWD resistance rank that each maize variety achieved in those tests. This comparative analysis illustrates…”.

Also, the following explanation was added to Table 3 footnotes (lines 455-456): “On the top are the highest-resistance cultivars and the least-resistant ones are at the bottom.”

Line 430, Sweet corn seedling (sprout) biomass is generally less than field corn due to the reduced carbohydrate content in the kernel.

Reply: Indeed. The above explanation was added to the text (lines 486-489): “Hence, it seems that the sprouts’ biomass is affected by factors that are not only related to the disease (such as the vitality of the seeds, their carbohydrate content and the genetic differences between them).”

Reviewer 2 Report

Cultivating disease-resistant maize varieties is the primary means for reducing late wilt disease (LWD)’s impact. This manuscript established a rapid laboratory seedling pathogenicity test and full-season open encloser semi-field conditioned pots assay to classify maize varieties according to their LWD resistance. I am confused about the data in Table 3. For example, the order in qPCR 50 DAS is not consistent with the data in Figure 5A. Maybe I did not understand the data processing method. Therefore, authors should check the data carefully in Table 3 (not only for qPCR 50 DAS) or indicate clearly how they get their conclusion from the data. In Figure 7, I wonder the relationship between the four disease grades and the data in Figure 7A and 7B. Authors should explain it in the Materials and Methods.

Author Response

Responses to Reviewer 2’s comments

We would like to express our sincere appreciation to the reviewer for the essential and helpful advice. The time and effort invested are greatly appreciated and certainly contributed to the manuscript and improved it. Thank you. All your remarks and suggestions were addressed carefully and thoroughly, as detailed below.

Cultivating disease-resistant maize varieties is the primary means of reducing late wilt disease (LWD)’s impact. This manuscript established a rapid laboratory seedling pathogenicity test and full-season open encloser semi-field conditioned pots assay to classify maize varieties according to their LWD resistance

Reply: Thank you for the positive evaluation of our manuscript. All your remarks and suggestions were addressed carefully and thoroughly, as detailed below.

I am confused about the data in Table 3. For example, the order in qPCR 50 DAS is not consistent with the data in Figure 5A. Maybe I did not understand the data processing method. Therefore, the authors should check the data carefully in Table 3 (not only for qPCR 50 DAS) or indicate clearly how they get their conclusion from the data.

Reply: The reviewer is correct. We apologize for this editing mistake. All the data in table 3 were double-checked and updated. The values were ranked using the Excel command “Rank.” The corn varieties are ordered in the total rank column according to their sum of all scores. This explanation was added to the text (lines 434-435, 460-461).

In Figure 7, I wonder about the relationship between the four disease grades and the data in Figures 7A and 7B. The authors should explain it in the Materials and Methods.

Reply: Thank you for this important remark. While the air photograph allows easy scanning of a large group of plants and identifies wilted plants and “hot spots of disease” in the field, this method is less accurate than evaluating the symptoms from the ground. On the overhand, symptoms’ evaluation from the ground requires investment in time and workforce and, if conducted by several persons, can be subjected to differences in individual judgment. We used both ways to demonstrate their advantages and weak points so that this information can be considered and assist in decision-making regarding the preferred evaluation method of choice in future works.

The above information was added to the Materials and Methods (lines 204-211).

Reviewer 3 Report

P 15, the “ 5. Conclusions”  M. maydis need italic.

Author Response

Responses to Reviewer 3’s comments

We thank the reviewer for investing substantial efforts, which undoubtedly contribute to this manuscript. Your contribution is greatly appreciated.

 P 15, the “ 5. Conclusions” M. maydis need italic.

Reply: The fungus name was corrected to italic. Thank you. We double-check the entire manuscript to ensure that the fungus name appears correctly.